# Baseline Variability Affects N-of-1 Intervention Effect: Simulation and Field Studies

**DOI:** 10.3390/jpm13050720

**Published:** 2023-04-24

**Authors:** Makoto Suzuki, Satoshi Tanaka, Kazuo Saito, Kilchoon Cho, Naoki Iso, Takuhiro Okabe, Takako Suzuki, Junichi Yamamoto

**Affiliations:** 1Faculty of Health Sciences, Tokyo Kasei University, 2-15-1 Inariyama, Sayama City 350-1398, Japan; 2Faculty of Systems Design, Tokyo Metropolitan University, 1-1 Minami-Osawa, Hachioji City 192-0397, Japan; 3Laboratory of Psychology, Hamamatsu University School of Medicine, 1-20-1 Handayama, Higashi-ku, Hamamatsu City 431-3192, Japan; 4School of Health Sciences, Saitama Prefectural University, 820 Sannomiya, Koshigaya City 343-8540, Japan

**Keywords:** local linear trend model, baseline-data variability, intervention effect, precision rehabilitation, N-of-1 trial

## Abstract

The simulation study investigated the relationship between the local linear trend model’s data-comparison accuracy, baseline-data variability, and changes in level and slope after introducing the N-of-1 intervention. Contour maps were constructed, which included baseline-data variability, change in level or slope, and percentage of non-overlapping data between the state and forecast values by the local linear trend model. Simulation results showed that baseline-data variability and changes in level and slope after intervention affect the data-comparison accuracy based on the local linear trend model. The field study investigated the intervention effects for actual field data using the local linear trend model, which confirmed 100% effectiveness of previous N-of-1 studies. These results imply that baseline-data variability affects the data-comparison accuracy using a local linear trend model, which could accurately predict the intervention effects. The local linear trend model may help assess the intervention effects of effective personalized interventions in precision rehabilitation.

## 1. Introduction

An N-of-1 research design can be used to examine the effectiveness of rehabilitation interventions for a single patient. This research design involves multiple measurements over time in one patient and is advantageous for assessing behavioral and functional changes in individual patients in various rehabilitation settings [1,2,3]. Two major types of N-of-1 research designs are reversal design (Figure 1A) and multiple-baseline design (Figure 1B) [4,5,6,7]. A reversal design runs consecutive sessions by alternating control and treatment conditions, while a multiple-baseline design runs consecutive sessions by allowing each control condition for a different number of data points.

Such designs expose the patient to both control (baseline phase) and treatment (intervention phase) conditions, thus comparing individuals’ behavior and function between the baseline and intervention phases to discover whether the behavior and function in the control condition are changed after treatment initiation [8,9,10]. A previous study noted that a meta-analysis of randomized N-of-1 research designs represented the highest level of evidence in clinical practice [11].

Once the N-of-1 data in the baseline and intervention phases are obtained, they are conventionally compared between the phases using various analysis methods, such as the percentage of non-overlapping data assuming a stable slope (PND-STB), the split-middle line, autoregressive integrated moving average model (ARIMA), and Tau-U [12,13,14,15]. However, interpretation of these conventional analysis methods is limited to over- and under-estimation because these methods assume a linearly stable slope of the data changes. However, the patient’s small and non-normal data would not be linearly but nonlinearly or unstably changed with time [16,17]. Recently performed Bayesian estimates, such as the local linear trend model (LLT), are expected to be a new, useful analysis method because they can assume both the unstable changing level and the slope of the data [18,19]. Therefore, this model can compare N-of-1 data with time-course changes in the level and slope between the baseline and intervention phases.

Although the LLT is useful for the analysis of small and non-normal N-of-1 data with time-course changes in level and slope, little is known about the accuracy of comparison between baseline and intervention phases for the LLT. A previous study noted that the quality of data in the baseline phase influences the data comparison between baseline and intervention phases: first, baseline data should be less variable and highly stable to accurately predict future data; second, baseline data should lack any trend to allow for accurate evaluation of the change in slope and level of the data after the introduction of the treatment [20,21]. Based on previous studies on the quality of baseline data, we hypothesized that the accuracy of data comparison between baseline and intervention phases may be affected by the variability of baseline data and changes in slope and level after treatment initiation. Correspondingly, the LLT may accurately predict the effectiveness of actual field data. Exploring how the accuracy of data comparison between baseline and intervention phases is affected by baseline-data variability and changes in level and slope after treatment using the LLT may contribute to our understanding of the N-of-1 research design to examine the effectiveness of rehabilitation intervention for a single patient.

To address these issues, the present study aimed (1) to compare data-comparison accuracy between the LLT and conventional PND-STB methods according to baseline-data variability and compare changes in level and slope of the baseline data after introducing the N-of-1 intervention, (2) to explore the range of baseline-data variability and changes in level and slope for acceptable data-comparison accuracy using a simulation study, and (3) to examine the effect of intervention using the LLT and PND-STB in an actual N-of-1 field study.

## 2. Material and Methods

### 2.1. Simulation Study

Our study was approved by the Research Ethics Committee of the Tokyo Kasei University (SKE2021-12) and followed the guidelines of the Declaration of Helsinki. We predicted that individual data, such as behavior and function, would randomly fluctuate and that the level and slope of the data would change after commencing treatment. Therefore, to generate the individual data, a simple state-changing model was constructed, which included the level as immediate effects, slope as retardative effects, and random fluctuation as data variability process, as follows:(1)$$f\left(t\right)=\alpha +\beta t+\varepsilon_{t}$$
where α is the y-intercept of the data, reflecting the level of the data as immediate effects; β is the slope of the data, reflecting changes in the data as retardative effects; $\varepsilon_{t}$ is the random variation, reflecting the fluctuation of the data as data variability; and t is the number of trials during the baseline and intervention phases. Figure 2 and Figure 3 show the representative simulation data of the change in the level and slope, respectively, derived from Equation (1). $\varepsilon_{t}$ deviated from the value of the slope trajectory by 0.0 and ±1.2 (0.2 steps) for the median value of the baseline data. α of the baseline phase was 1.0, whereas α of the intervention phase multiplied the level of the end of the baseline data by 1.0 to 1.6 (0.1 steps), reflecting the change in level. β of the baseline phase was 0.1, whereas β of the intervention phase multiplied the slope of the baseline phase by 1.0 to 2.2 (0.2 steps), reflecting the change in slope. t was set from 0 to 4 in the baseline phase and from 5 to 14 in the intervention phase in accordance with the number of trials in previous N-of-1 research [2,3,7,10,15]. Equations were constructed using the Python 3.9.7 (Python Software Foundation, Wilmington, DE, USA).

Subsequently, the LLT was applied to the simulated data. The LLT assumes that both levels (Equation (3)) and slope (Equation (4)) from the simulation data (Equation (2)) follow Gaussian random walks. An LLT was constructed for the simulated data as follows:(2)$$y_{t}=\mu_{t}+\varepsilon_{t}$$
(3)$$\mu_{t+1}=\mu_{t}+\nu_{t}+\xi_{t}$$
(4)$$\nu_{t+1}=\nu_{t}+\zeta_{t}$$
where $y_{t}$ is the simulation data, $\varepsilon_{t}$ is the random variable, $\mu_{t}$ is the level, $\nu_{t}$ is the slope, $\xi_{t}$ is the disturbance at the level, and $\zeta_{t}$ is the disturbance at the slope [18,19]. The LLT estimated the state value with random variables ($\varepsilon_{t}$) in the baseline and intervention phases (state value) and forecasted the value by extrapolating the baseline value to the intervention phase using the level ($\mu_{t}$) and slope ($\nu_{t}$) (forecast value). After calculating state and forecast values, the values in the intervention phase were compared using the percentage of non-overlapping data between state and forecast values (PND-LLT) and PND-STB, as follows:(5)$$\mathrm{PND}-\mathrm{LLT}\;\mathrm{or}-\mathrm{STB}=100\frac{g}{n}$$
where g is the number of state values greater than forecast values (PND-LLT) or the number of raw data in the intervention phase greater than the maximal data in the baseline phase (PND-STB), and n is the number of sessions in the intervention phase. In current practice, PND values in the range of 90–100 are accepted as “very effective”, those in the range of 70–90 as “effective”, those in the range of 50–70 as “questionable”, and those below 50 as “ineffective” [22]. The generation and comparison of simulated data were repeated 100 times, and the PND-LLT and conventional PND-STB were calculated in each step of random error and changes in level and slope. The LLT was conducted using the statsmodels package in the Python environment.

Next, we predicted that the variability of baseline data ($\varepsilon_{t}$ of Equation (1)) and changes in the level (α of Equation (1)) and slope (β of Equation (1)) after introducing treatment would affect the data-comparison accuracy between the state and forecast values because the variability of baseline data with trend influences the data comparison [20,21]. Therefore, contour maps were constructed, which included baseline-data variability, change in level or slope, and the PND-LLT or conventional PND-STB.

In addition, the PND for the $\varepsilon_{t}$ values of 0.0 ± 0.0, reflecting true value without random variation; the PND for 1.0 changes in the level and slope, reflecting no-intervention effects; and the PND for 1.6 changes in the level and slope, reflecting maximal intervention effects, were extracted from the PND-LLT and PND-STB.

Furthermore, to contribute to the clinical indices, the coefficient of variation (CV) values for $\varepsilon_{t}$ values of 0.0–1.2 were calculated as well as the range of baseline-data variability (i.e., CV values) and changes in level and slope for acceptable data-comparison accuracy (i.e., PND value ≥ 70).

### 2.2. Field Study

In addition, field testing involved applying PND-LLT and PND-STB to 17 N-of-1 trial datasets from the behavioral rehabilitation database, based on published Japanese articles [23,24,25,26,27,28,29,30,31,32,33,34,35,36,37,38,39] (Table 1). Eligibility criteria for field studies included an N-of-1 research design, rehabilitation training, and a period of more than 2 and 4 days in baseline and intervention phases, respectively. All participants provided informed consent prior to participation in each published article. State and forecast values were calculated using the LLT, and the values in the intervention phase were compared using the PND-LLT and the PND-STB (Equation (5)). In the present study, when the PND value was ≥ 70, the intervention was considered effective.

## 3. Results

### 3.1. Simulation Study

Baseline-CV values for $\varepsilon_{t}$ values of 0.0, 0.2, 0.4, 0.6, 0.8, 1.0, and 1.2 are shown in Table 2. Table 3 and Figure 4 show the PND-LLT between the state and forecast values (Equation (5)) based on simulation data (Equation (1)). In the PND-LLT, the PND values increased in accordance with the increment of change in the level (i.e., immediate effects) or slope (i.e., retardative effects). However, the PND values decreased according to the increment of baseline variabilities for 1.1 to 1.6 changes in the level (i.e., increasing type II error), whereas the PND values increased according to the increment of baseline variabilities for 1.0 changes in the level (i.e., increasing type I error). Similarly, the PND values decreased in accordance with the increment of baseline variabilities for 1.2 to 2.2 changes in the slope (i.e., increasing type II error), whereas the PND values increased with the increment of baseline variabilities for 1.0 changes in the slope (i.e., increasing type I error). Additionally, the ranges of baseline-CV values and changes in level and slope for acceptable data-comparison accuracy (i.e., PND value ≥ 70) are denoted with asterisks in Table 3. For the PND-LLT, the limitations of baseline-CV values for detectable changes in level were 0.13 ± 0.00 (Error 0.0) for a change in level of 1.1, 0.17 ± 0.01 (Error 0.2) for a change of 1.3, and 0.26 ± 0.01 (Error 0.4) for 1.5, and CV values ≥ 0.36 ± 0.01 (Error 0.6) could not detect changes in levels of 1.1–1.6. Likewise, the limitations of baseline-CV values for detectable changes in slope were 0.13 ± 0.00 (Error 0.0) for a change in slope of 1.2 and 0.17 ± 0.01 (Error 0.2) for a change of 2.0, and CV values ≥ 0.25 ± 0.01 (Error 0.4) could not detect changes in slope of 1.2–2.2.

Table 4 and Figure 5 show the PND-STB between raw data in the intervention phase and maximal data in the baseline phase (Equation (5)) based on simulation data (Equation (1)). In the PND-STB, the PND values increased in accordance with the increment of change in the level or slope. However, the PND values decreased according to the increment of baseline variabilities for 1.1 to 1.6 changes in the level (i.e., increasing type II error). Similarly, the PND values decreased in accordance with the increment of baseline variabilities for 1.2 to 2.2 changes in the slope (i.e., increasing type II error). Notably, the PND values were high regardless of the baseline variabilities for 1.0 changes in the level and slope with Error 0.0 (i.e., high type I error). Additionally, for the PND-STB, the limitations of baseline-CV values for detectable changes in level were 0.36 ± 0.01 (Error 0.6) for a change in level of 1.1, 0.48 ± 0.02 (Error 0.8) for 1.3, 0.58 ± 0.02 (Error 1.0) for 1.4, and 0.80 ± 0.04 (Error 1.2) for 1.6. Likewise, the limitations of baseline-CV values for detectable changes in slopes were 0.25 ± 0.01 (Error 0.4) for a change in slope of 1.2, 0.36 ± 0.01 (Error 0.6) for a change of 1.2, and 0.48 ± 0.02 (Error 0.8) for 1.6, and the CV value of 0.59 ± 0.02 (Error 1.0) could not detect changes in slope of 1.2–2.2. However, the PND-STB totally overestimated the changes in level and slope (i.e., high type I error).

Furthermore, the PND for the Error 0.0 (i.e., y-axis of the contour maps in Figure 4 and Figure 5), reflecting true value without random variation; PND for 1.0 changes in the level and slope (i.e., x-axis of the contour maps in Figure 4 and Figure 5), reflecting no-intervention effects; and PND for 1.6 changes in the level and slope (i.e., upper limit of the contour maps in Figure 4 and Figure 5), reflecting maximal intervention effects, were extracted from the PND-LLT and PND-STB (Figure 6). In the Error 0.0 (Figure 6A,D), the PND values increased in accordance with the increment of change in the level or slope for the PND-LLT, whereas the PND values were consistently 100 for the mean PND-STB. Specifically, the PND value was zero at 1.0 change in the level and slope (i.e., low type I error) for the PND-LLT, whereas the PND values were 100 at 1.0 changes in the level and slope (i.e., high type I error) for the PND-STB. At the 1.0 change in the level and slope (Figure 6B,E), the PND values increased in accordance with the increment of baseline variabilities (i.e., increasing type I error) for the PND-LLT, while the PND values decreased according to the increment of baseline variabilities for the PND-STB to the same level as the PND-LLT. In the 1.6 changes in the level and slope (Figure 6C,F), the PND values decreased in accordance with the increment of baseline variabilities at 1.6 changes in the level (i.e., increasing type II error) for the PND-LLT and PND-STB.

### 3.2. Field Study

The effectiveness of intervention in each of the 17 N-of-1 studies was re-examined using the PND-LLT and the PND-STB. Participants in our field study were patients over the age of 40 years with various diseases, such as brain injury, stroke, dementia, and cervical myelopathy. In addition, the field study included various target behaviors, such as dressing, using chopsticks, and walking, and various interventions, such as token economy, shaping, and reinforcement. Figure 7 presents the results of the field study for the actual N-of-1 trial datasets. For all 17 (100%) studies, the PND-LLT values were ≥70. However, for 15 of the 17 (88%) studies, the PND-STB values were ≥70. Additionally, for 8 of the 17 (47%) studies, the PND-LLT values were higher than the PND-STB values.

## 4. Discussion

This study aimed to test the hypothesis that the accuracy of data comparison between baseline and intervention phases is affected by the variability of baseline data and changes in slope and level after treatment initiation, and that the LLT can accurately predict the effectiveness of actual field data. Our computational simulation results showed that baseline-data variability and changes in the level and slope after the intervention affect the accuracy of data comparison based on the PND-LLT. In addition, according to our field results, the PND-LLT confirmed the effectiveness of 100% of previous N-of-1 studies in actual field intervention.

The first novel observation of our study is that the baseline-data variability and changes in the level and slope affect the accuracy of data comparison based on the PND-LLT. Once the N-of-1 data in the baseline and intervention phases were obtained, they were conventionally compared between the phases using the PND-STB [12,40,41]. In the PND-STB, the percentage of raw data in the intervention phase greater than the maximal data in the baseline phase was calculated [12,40,41]. Although conventional analysis methods, such as the PND-STB, are useful for N-of-1 data analysis, they cannot assume a change in the slope of the data, apart from in linearly stable data. Therefore, these analyses cannot precisely predict the current and future data. In our study, the PND values between the state and forecast values increased in accordance with the increment of baseline-variabilities for 1.0 changes in the level or slope for the PND-LLT, although the PND increased in response to an increment of more than a 1.0 change in level or slope. However, the PND values were high regardless of the baseline variabilities for 1.0 changes in the level and slope in the PND-STB. These simulation results indicate that baseline-data variability is important for understanding the effectiveness of an intervention and that the PND-LLT could more accurately predict the intervention effects compared to conventional PND-STB because it had fewer type I errors. These observations may help understand and promote personalized interventions based on an N-of-1 research design from the perspective of each individual’s precision rehabilitation.

We calculated the CV values in the baseline phase and explored the range of baseline-CV values and changes in level and slope for acceptable data-comparison accuracy. For the PND-LLT, the limitations of baseline-CV values for detectable changes in level were 0.13 ± 0.00, 0.17 ± 0.01, or 0.26 ± 0.01 for changes in level of 1.1, 1.3, or 1.5, respectively. Likewise, the limitations of baseline-CV values for detectable changes in slope were 0.13 ± 0.00 or 0.17 ± 0.01 for changes in slope of 1.2 or 2.0, respectively. However, the baseline-CV value of ≥0.36 ± 0.01 or 0.26 ± 0.01 could not detect changes in level of 1.1–1.6 or the change in slope of 1.2–2.2, respectively. Our results suggest that changes in level and slope for detecting the intervention effect can be estimated by an assessment of baseline-CV values. Therefore, the ranges of baseline-CV values and changes in level and slope for acceptable data-comparison accuracy may contribute to an increasingly evidence-based personalized approach in clinical rehabilitation settings. Future research should address baseline-CV variability ≥ 0.36 ± 0.01 or 0.26 ± 0.01 and the detection of changes in level and slope.

The second novel finding of our study is that PND-LLT predicted that all 17 (100%) field studies conducted effective interventions. In fact, in 8 of 17 (47%) studies, the PND-LLT values were higher than the PND-STB values. These field results indicate that the PND-LLT could more sensitively predict the intervention effects compared to the PND-STB. In the original studies, the N-of-1 data were conventionally compared between the baseline and intervention phases using not only the PND-STB but also visual inspection, split-middle line, ARIMA, or Tau-U [12,13,14,15,42]. In visual inspection, the trend and level of N-of-1 data are visually compared between phases without mathematical operations [42]. ARIMA can assess the difference between current and previous values with autoregression and can forecast future values [43]. Furthermore, Tau-U can consider the trend of the baseline and intervention phases and compare the data between the two [15]. These conventional analysis methods assume a linearly stable slope of the data, similar to the PND-STB. However, we only focused on the data-comparison accuracy between the PND-LLT and the conventional PND-STB. Therefore, future studies are needed to clarify the data-comparison accuracy for other analysis methods, including PND-LLT, the split-middle line, ARIMA, and Tau-U. When performing a rehabilitation intervention, therapists assess the effectiveness of the treatment using an N-of-1 paradigm and modify the treatment according to the analysis results based on N-of-1 research. These objective assessments may minimize intervention bias, such as continuation of less effective interventions. The PND-LLT applied to actual field data may offer advantages in future personalized interventions from the perspective of precision rehabilitation, as analysis reaching scientific significance may help clinicians to determine whether to continue the intervention.

Our study had several limitations. First, the number of trials in the baseline and intervention phases was set to 5 and 10, respectively, based on the number of sessions in previous N-of-1 research [2,3,7,10,15]. Our simulation results could not be applied to much longer or shorter trial periods, although a small number of trials (i.e., 5 to 10) in the baseline and intervention phases could be applied to broad rehabilitation interventions. Therefore, further research is needed to explore the relationship between the data-comparison accuracy of the PND-LLT, baseline-data variability, and changes in the slope and level after intervention using a simulation study with various trial periods. Second, our field study used non-probability samples. This sampling method contains some bias compared to random selection, because the N-of-1 trials do not have an equal chance of being selected under the specific inclusion criteria. Thus, larger probability samples are needed in future studies to enable us to readily assume that the sample represents a broad population in the case of rehabilitation interventions.

## 5. Conclusions

In conclusion, through a simulation study, we found that baseline-data variability and changes in level and slope after N-of-1 intervention affect the accuracy of data comparison based on the PND-LLT. In addition, the PND-LLT could accurately predict the intervention effects for actual data in a field analysis. Therefore, the PND-LLT may serve as a sensitively meaningful analysis method in personalized clinical practice that uses an N-of-1 research design. However, large baseline-data variability (the CV values of ≥0.36 ± 0.01 or 0.26 ± 0.01) could not detect all changes in level or slope, respectively. These findings contribute toward an increasingly evidence-based approach for the personalized rehabilitation of individual patients.

## Figures and Tables

**Figure 1 jpm-13-00720-f001:**
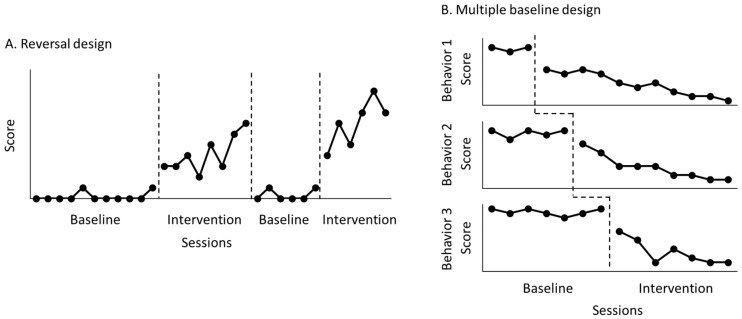
Reversal design (**A**) and multiple-baseline design (**B**). In the reversal design, the researcher monitors a behavior by alternating control with treatment conditions. In the multiple-baseline design, the researcher monitors several behaviors by using a different number of control data points under the same treatment.

**Figure 2 jpm-13-00720-f002:**
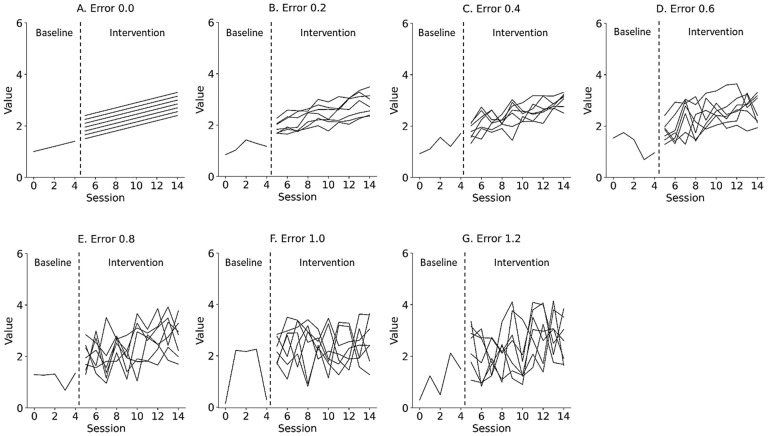
Representative simulation data with random errors of 0.0 (**A**), 0.2 (**B**), 0.4 (**C**), 0.6 (**D**), 0.8 (**E**), 1.0 (**F**), and 1.2 (**G**) derived from a simple state-changing model. Lines denote changes in level, ranging from 1.0 to 1.6 (0.1 step). Trials 0 to 4 denote baseline phase and 5 to 14 denote intervention phase. Errors 0.0–1.2 denote $\varepsilon_{t}$ values 0.0 ± 0.0 (Error 0.0), 0.0 ± 0.2 (Error 0.2), 0.0 ± 0.4 (Error 0.4), 0.0 ± 0.6 (Error 0.6), 0.0 ± 0.8 (Error 0.8), 0.0 ± 1.0 (Error 1.0), and 0.0 ± 1.2 (Error 1.2) for the median value of the baseline data.

**Figure 3 jpm-13-00720-f003:**
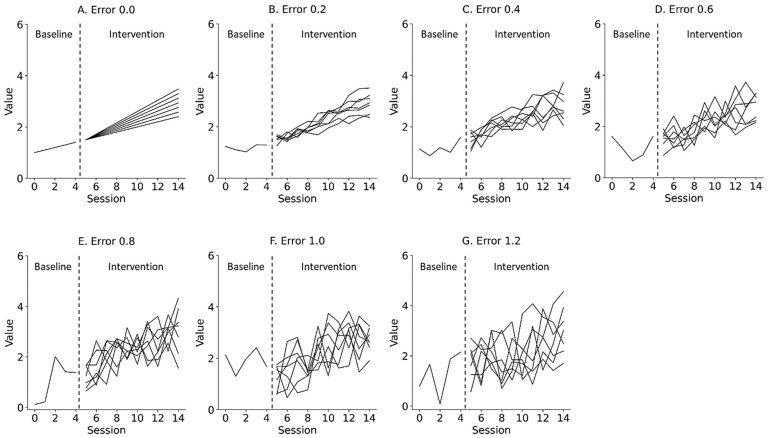
Representative simulation data with random errors of 0.0 (**A**), 0.2 (**B**), 0.4 (**C**), 0.6 (**D**), 0.8 (**E**), 1.0 (**F**), and 1.2 (**G**) derived from a simple state-changing model. Lines denote changes in slope, ranging from 1.0 to 2.2 (0.2 step). Trials 0 to 4 denote baseline phase, and 5 to 14 denote intervention phase. Errors 0.0–1.2 denote $\varepsilon_{t}$ values 0.0 ± 0.0 (Error 0.0), 0.0 ± 0.2 (Error 0.2), 0.0 ± 0.4 (Error 0.4), 0.0 ± 0.6 (Error 0.6), 0.0 ± 0.8 (Error 0.8), 0.0 ± 1.0 (Error 1.0), and 0.0 ± 1.2 (Error 1.2) for the median value of the baseline data.

**Figure 4 jpm-13-00720-f004:**
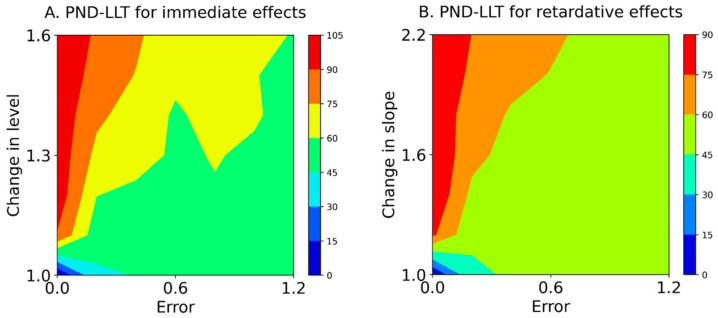
Mean contours of the PND-LLT for immediate effects (**A**) and retardative effects (**B**). The color of each contour denotes the PND range. PND-LLT: percentage of non-overlapping data between state and forecast values. Errors 0.0–1.2 denote $\varepsilon_{t}$ values 0.0 ± 0.0 (Error 0.0), 0.0 ± 0.2 (Error 0.2), 0.0 ± 0.4 (Error 0.4), 0.0 ± 0.6 (Error 0.6), 0.0 ± 0.8 (Error 0.8), 0.0 ± 1.0 (Error 1.0), and 0.0 ± 1.2 (Error 1.2) for the median value of the baseline data.

**Figure 5 jpm-13-00720-f005:**
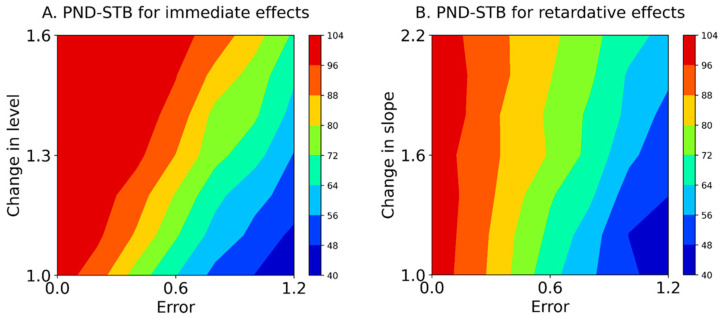
Mean contours of the PND-STB for immediate effects (A) and retardative effects (B). The color of each contour denotes the PND range. PND-STB: the percentage of non-overlapping data assuming a stable slope. Errors 0.0–1.2 denote $\varepsilon_{t}$ values 0.0 ± 0.0 (Error 0.0), 0.0 ± 0.2 (Error 0.2), 0.0 ± 0.4 (Error 0.4), 0.0 ± 0.6 (Error 0.6), 0.0 ± 0.8 (Error 0.8), 0.0 ± 1.0 (Error 1.0), and 0.0 ± 1.2 (Error 1.2) for the median value of the baseline data.

**Figure 6 jpm-13-00720-f006:**
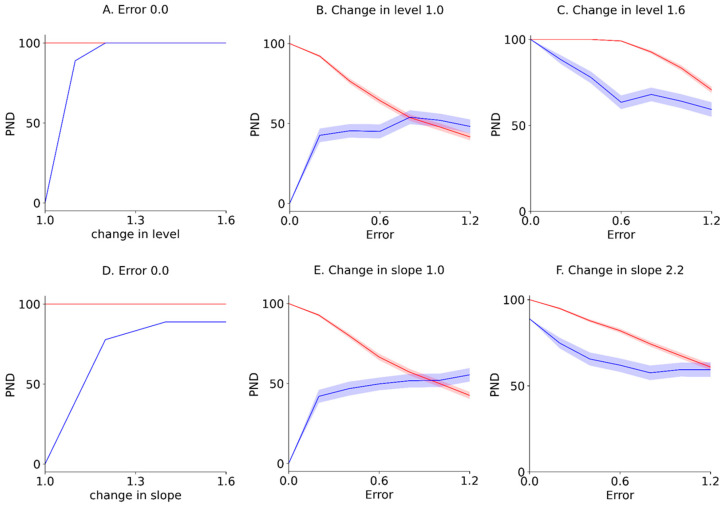
Extracted mean ± standard error of the mean PND for Error 0.0 (**A**,**D**), those for 1.0 changes in the level and slope (**B**,**E**), and those for 1.6 changes in the level and slope (**C**,**F**). The blue and red solid lines indicate the mean PND-LLT and -STB, respectively. The colored areas indicate the standard error of the mean. PND: percentage of non-overlapping data. Errors 0.0–1.2 denote $\varepsilon_{t}$ values 0.0 ± 0.0 (Error 0.0), 0.0 ± 0.2 (Error 0.2), 0.0 ± 0.4 (Error 0.4), 0.0 ± 0.6 (Error 0.6), 0.0 ± 0.8 (Error 0.8), 0.0 ± 1.0 (Error 1.0), and 0.0 ± 1.2 (Error 1.2) for the median value of the baseline data.

**Figure 7 jpm-13-00720-f007:**
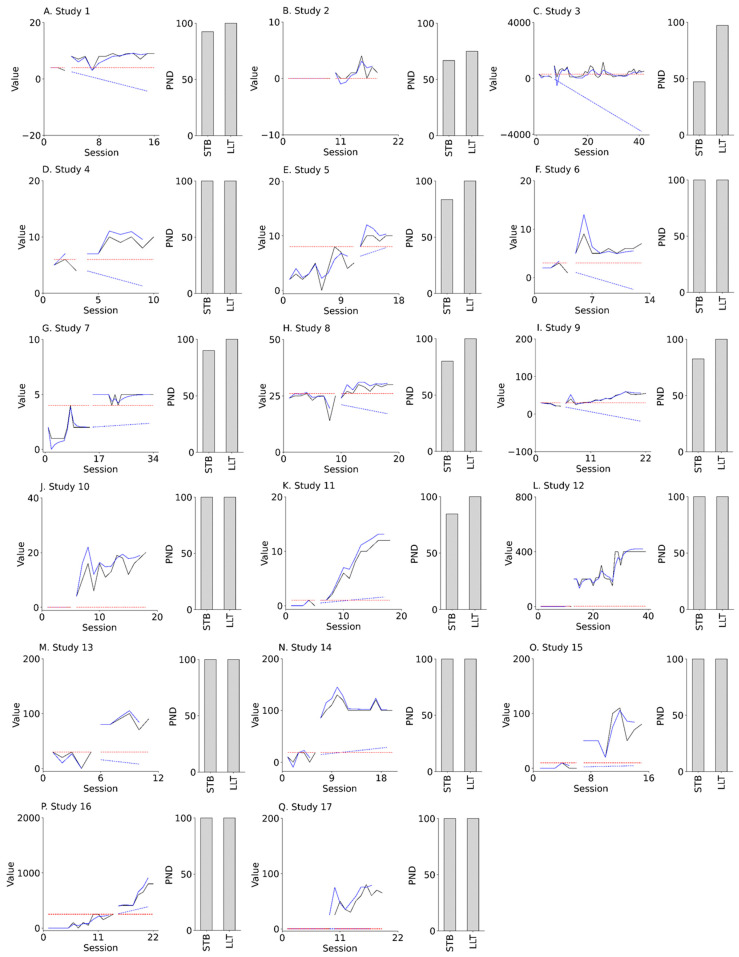
Actual field data (left panels of **A**–**Q**) and PND values (right panels of **A**–**Q**). Black solid lines denote actual data. Blue solid lines and dotted lines denote state value and forecast value, respectively, derived from LLT. Red dotted lines denote maximal data in the baseline phase. PND: percentage of non-overlapping data; STB: stable slope; LLT: local linear trend.

**Table 1 jpm-13-00720-t001:** Field data from the behavioral rehabilitation database based on published Japanese N-of-1 articles.

Study	Diagnosis	Age	Target Behavior	Intervention	Design	Analysis
1	Brain injury	43	Dressing	Token economy	AB	SML
2	Stroke	44	Using chopsticks	Shaping	ABAB	SML
3	Dementia	71	Walking	Reinforcement	AB	SML
4	Stroke	55	Bathing	Shaping	AB	SML
5	Stroke	64	Walking	Token economy	ABAB	SML
6	Stroke	70	Selfcare	Shaping	ABAB	SML
7	Dementia	82	Standing up	Shaping	AB	VI
8	Cervical myelopathy	66	Getting up	Shaping	ABAB	VI
9	Supranuclear palsy	80	Getting up	Shaping	AB	SML
10	Dementia	80	Wheelchair operation	Shaping	AB	VI
11	Stroke	78	Getting up	Shaping	AB	VI
12	Stroke	60	Standing up	Reinforcement	AB	VI
13	Stroke	80	Talking	Reinforcement	ABAB	VI
14	Spinal cord injury	80	Walking	Reinforcement	AB	VI
15	Dementia	80	Training adherence	Reinforcement	AB	VI
16	Stroke	70	Training adherence	Reinforcement	AB	VI
17	Stroke	71	Talking	Reinforcement	ABAB	VI

AB: AB design; ABAB: reversal design; SML: split-middle line; VI: visual inspection.

**Table 2 jpm-13-00720-t002:** Baseline coefficient of variation values for random variations ($\varepsilon_{t}$ ) of the model.

**A. Change in Level**							
	**Random Variation in Baseline Phase**						
$\varepsilon_{t}$	0.0	0.2	0.4	0.6	0.8	1.0	1.2
CV	0.13 ± 0.00	0.17 ± 0.01	0.26 ± 0.01	0.36 ± 0.01	0.48 ± 0.02	0.58 ± 0.02	0.80 ± 0.04
**B. Change in Slope**							
	**Random Variation in Baseline Phase**						
$\varepsilon_{t}$	0.0	0.2	0.4	0.6	0.8	1.0	1.2
CV	0.13 ± 0.00	0.17 ± 0.01	0.25 ± 0.01	0.36 ± 0.01	0.48 ± 0.02	0.59 ± 0.02	0.73 ± 0.04

Values are presented as mean ± standard error of the mean. CV: coefficient of variation.

**Table 3 jpm-13-00720-t003:** The percentage of non-overlapping data with local linear trend.

		Error in Baseline Phase						
		0.0	0.2	0.4	0.6	0.8	1.0	1.2
Change in level	1.0	0 ± 0	43 ± 4	45 ± 4	45 ± 4	54 ± 4	52 ± 4	48 ± 4
	1.1	89 ± 0 *	51 ± 4	53 ± 4	49 ± 4	55 ± 4	53 ± 4	52 ± 4
	1.2	100 ± 0 *	60 ± 4	57 ± 4	52 ± 4	59 ± 4	54 ± 4	53 ± 4
	1.3	100 ± 0 *	71 ± 4 *	65 ± 4	58 ± 4	61 ± 4	58 ± 4	54 ± 4
	1.4	100 ± 0 *	78 ± 3 *	68 ± 4	58 ± 4	64 ± 4	61 ± 4	56 ± 4
	1.5	100 ± 0 *	85 ± 3 *	74 ± 4 *	63 ± 4	65 ± 4	60 ± 4	57 ± 4
	1.6	100 ± 0 *	88 ± 3 *	78 ± 3 *	63 ± 4	68 ± 4	64 ± 4	59 ± 4
Change in slope	1.0	0 ± 0	42 ± 4	47 ± 4	50 ± 4	52 ± 4	52 ± 4	55 ± 4
	1.2	78 ± 0 *	48 ± 4	49 ± 4	48 ± 4	52 ± 4	51 ± 4	53 ± 4
	1.4	89 ± 0 *	56 ± 4	53 ± 4	54 ± 4	52 ± 4	53 ± 4	55 ± 4
	1.6	89 ± 0 *	65 ± 4	55 ± 4	56 ± 4	55 ± 4	54 ± 4	57 ± 4
	1.8	89 ± 0 *	66 ± 4	59 ± 4	59 ± 4	54 ± 4	56 ± 4	57 ± 4
	2.0	89 ± 0 *	72 ± 3 *	63 ± 4	60 ± 4	60 ± 4	56 ± 4	58 ± 4
	2.2	89 ± 0 *	75 ± 3 *	66 ± 4	62 ± 4	58 ± 4	59 ± 4	59 ± 4

Values are expressed as mean ± standard error of the mean. Errors 0.0–1.2 denote $\varepsilon_{t}$ values 0.0 ± 0.0 (Error 0.0), 0.0 ± 0.2 (Error 0.2), 0.0 ± 0.4 (Error 0.4), 0.0 ± 0.6 (Error 0.6), 0.0 ± 0.8 (Error 0.8), 0.0 ± 1.0 (Error 1.0), and 0.0 ± 1.2 (Error 1.2) for the median value of the baseline data. *: percentage of non-overlapping data ≥ 70.

**Table 4 jpm-13-00720-t004:** The percentage of non-overlapping data with a stable slope.

		Error in Baseline Phase						
		0.0	0.2	0.4	0.6	0.8	1.0	1.2
Change in level	1.0	100 ± 0	92 ± 1	76 ± 2	64 ± 2	54 ± 2	48 ± 2	42 ± 2
	1.1	100 ± 0 *	98 ± 0 *	87 ± 1 *	73 ± 2 *	61 ± 2	54 ± 2	46 ± 2
	1.2	100 ± 0 *	100 ± 0 *	92 ± 1 *	79 ± 2 *	66 ± 2	58 ± 2	53 ± 2
	1.3	100 ± 0 *	100 ± 0 *	98 ± 1 *	88 ± 1 *	75 ± 2 *	67 ± 2	55 ± 2
	1.4	100 ± 0 *	100 ± 0 *	100 ± 0 *	94 ± 1 *	77 ± 2 *	73 ± 2 *	61 ± 2
	1.5	100 ± 0 *	100 ± 0 *	100 ± 0 *	96 ± 1 *	86 ± 1 *	77 ± 2 *	65 ± 2
	1.6	100 ± 0 *	100 ± 0 *	100 ± 0 *	99 ± 0 *	93 ± 1 *	83 ± 2 *	71 ± 2 *
Change in slope	1.0	100 ± 0	93 ± 1	80 ± 1	67 ± 2	57 ± 2	50 ± 2	43 ± 2
	1.2	100 ± 0 *	94 ± 1 *	81 ± 1 *	70 ± 2 *	60 ± 2	48 ± 2	44 ± 2
	1.4	100 ± 0 *	94 ± 1 *	84 ± 1 *	73 ± 2 *	66 ± 2	52 ± 2	48 ± 2
	1.6	100 ± 0 *	94 ± 1 *	86 ± 1 *	79 ± 1 *	70 ± 2 *	58 ± 2	50 ± 2
	1.8	100 ± 0 *	95 ± 1 *	85 ± 1 *	79 ± 1 *	71 ± 2 *	62 ± 2	53 ± 2
	2.0	100 ± 0 *	96 ± 1 *	88 ± 1 *	80 ± 1 *	74 ± 1 *	63 ± 2	58 ± 2
	2.2	100 ± 0 *	95 ± 1 *	88 ± 1 *	82 ± 1 *	74 ± 2 *	68 ± 2	61 ± 2

Values are expressed as mean ± standard error of the mean. Errors 0.0–1.2 denote $\varepsilon_{t}$ values 0.0 ± 0.0 (Error 0.0), 0.0 ± 0.2 (Error 0.2), 0.0 ± 0.4 (Error 0.4), 0.0 ± 0.6 (Error 0.6), 0.0 ± 0.8 (Error 0.8), 0.0 ± 1.0 (Error 1.0), and 0.0 ± 1.2 (Error 1.2) for the median value of the baseline data. *: percentage of non-overlapping data ≥ 70.

## Data Availability

Derived data supporting the findings of this study are available from the corresponding author, M.S.

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
