# Peer review of "Baseline Variability Affects N-of-1 Intervention Effect: Simulation and Field Studies"

_jpm, 2023, doi:10.3390/jpm13050720_

Round 1

Reviewer 1 Report

Baseline variability affects N of 1 intervention effect: Bayesian simulation and field studies

This manuscript investigates the relationship between the local linear model’s data comparison accuracy and changes in level and slopes. However, there are several issues that need to be addressed,

1.      On page 1 in Introduction section, it would be helpful to briefly introduce that is N of 1 research design, for some readers who are not familiar to it

2.      In all the models with random variation, what is the specific distribution assumption for the random variation, and how the parameters in the distribution assumption influences the inference accuracy?

3.      In Figure 1, it’s not clear the meaning of “Error 0.0”, “Error 0.2”, etc, what does the numbers 0.0, 0.2 mean?

4.      Add more details about how each model was implemented using software, e.g. what R packages or functions are used?

Reviewer 2 Report

The authors presented a simulation study to assess the accuracy of Bayesian methods to address the issue of non-linearity in data changes between baseline and intervention phase in N of 1 design.   It is very hard to understand the point of this study. The authors mention a "Bayesian simulation" in the title, but it is not clear at what extent they carried out Bayesian simulation. On the opposite, it seems that least-square method was used The main drawback in using local linear trend for forecasting is that the variability of prediction is quite huge. So, on my point of view is quite trivial that the variability at baseline measurements and change after intervention, affect the data-comparison accuracy based on the local linear trend model. The local linear trend model tends to overfit and thus it does not provide a good prediction model, especially when there is a lot of noise. So I do not understand the real point of this study, since I am not aware of n-of-1 studies analyzed with local linear trends

Round 2

Reviewer 1 Report

The authors have addressed all my concerns. 

Reviewer 2 Report

I appreciate the extensive work the authors did to revise the manuscript. However, even after the review, understanding the point of this study still remains hard.   Regarding the references in the introduction, it's essential that they match the relevance of the problem and be up-to-date. It's a good rule of thumb to have at least 50% of the references in the last 3 to 5 years to ensure that the literature review is current. Furthermore, it would be helpful if the authors addressed the comment regarding how the conclusions of the study impact the approaches used to analyze N-of-1 trials. Providing a clear answer to this question would improve the paper's contribution to the field and make it more relevant to researchers and practitioners in the area.
